# Nailing of Layers: A Promising Way to Reinforce Concrete 3D Printing Structures

**DOI:** 10.3390/ma13071518

**Published:** 2020-03-26

**Authors:** A. Perrot, Y. Jacquet, D. Rangeard, E. Courteille, M. Sonebi

**Affiliations:** 1Univ. Bretagne Sud, UMR CNRS 6027, IRDL, F-56100 Lorient, France; yohan.jacquet@univ-ubs.fr; 2INSA Rennes, EA 3913, LGCGM, F-35000 Rennes, France; damien.rangeard@insa-rennes.fr (D.R.); eric.courteille@insa-rennes.fr (E.C.); 3School of Natural and Built Environment, Queen’s University Belfast, Belfast B17 1NN, UK; m.sonebi@qub.ac.uk

**Keywords:** cement-based materials, rheology, 3D printing, additive manufacturing, reinforcement

## Abstract

Today, the extrusion-based 3D printing of concrete is a potential breakthrough technology for the construction industry. It is expected that 3D printing will reduce the cost of construction of civil engineering structures (removal of formwork) and lead to a significant reduction in time and improve working environment conditions. Following the use of this additive manufacturing layer-wise process, it is required to change the way concrete structures are designed and reinforced, especially for the parts of the structure under tension loads. Indeed, the extrusion-based concrete 3D printing process does not allow for the production of conventional reinforced concrete, and there is a need to develop other ways of compensating for the low mechanical performances of concrete, particularly in tension. In this study, the reinforcement of printed structures by using steel nails through the deposited layers of fresh concrete was investigated. Additionally, three-layer and 10-layer samples were reinforced with nails with varying inclination and spacing. The results show that inclined nails can be used to provide a flexural strengthening of the printing material in different directions.

## 1. Introduction

Digital concrete is a major challenge which can increase production rates while reducing construction environmental impact, enlarge the architectural possibilities and improve the quality of built concrete structures [1,2,3,4,5]. Among digital concrete methods, extrusion-based 3D printing is probably the most developed technique. Recent research on 3D printing has almost all been focused on mix-design, rheological and process related issues [6,7,8]. It has allowed the production of a physically-based background in order to formulate concrete with the required fresh properties, and allowed us to evaluate a time window during which it is possible to deposit a new layer of cement-based material. Nowadays, some technical solutions have emerged in the development of successful concrete printing, and researchers have started to work on the structural performances of reinforced and unreinforced concrete printed structures [9,10,11,12,13,14,15,16].

The structural design of printed structures can take advantages of the shape freedom brought by the printing process, target “full compression” mode [17] and use topological optimization principles [18,19,20,21]. Topological optimization consists in a mathematical design step that optimizes material shape within a given design space, for a given set of loads and boundary conditions. For cement-based or other brittle and low tensile strength materials, it results in the design of double-curved walls or structures that minimize tensile stresses. However, some printed real-scale structures of housing, bridges and others require the addition of reinforcements in order to provide the material with sufficient tensile and flexural strengths [10]. Moreover, this is currently the only solution to comply with structural design standards.

Different solutions of reinforcement have been investigated by several researchers [22,23,24,25,26,27,28,29]. The first one was to add steel reinforcements or cables in a reserved zone after concrete printing [9,22]. In this case, the structures can be designed and worked as a conventional pre-stressed or reinforced concrete. It can be observed that at the present time, the placing of the steel bars is not automated, even if some projects tend to use a simultaneous printing of concrete and steel [23]. Using fibers is also a common method to reinforce printed concrete structures [12,13,24,25,26,27,28]. For example, steel [24], bio-based fibres [16,26], polymeric fibres [12,24], basalt fibres [28] or glass fibers [14] have already been tested to enhance the tensile behavior and the ductility of 3D printed cementitious materials. Bos et al. also studied the effect of a steel wire embedded in the deposited layer as a potential efficient reinforcement strategy [29]. However, those two previous methods did not provide reinforcement that crossed the interface between layers that were expected to be a potential weakness zone of the printed structures [11,30,31,32].

In this paper, a new strategy of reinforcement was investigated. This strategy consisted of the placement of nails through several layers after their deposit. The aim was to provide ductility and tensile and shear strength while giving additional strength to the interface between layers once the material hardened. This strategy can be easily automated using a robotic placement of the nail which can be a real advantage and beneficial in the context of digital construction.

The conditions of nail penetration in the fresh cement-based materials is first studied from a theoretical point of view taking into account the gradient of the mechanical properties through the height of the printed structures. This investigation helped to compute the maximum numbers of layers that can be nailed in one injection.

Additionally, the reinforcement effects by nails are tested using three point flexural tests. The orientation, surface roughness and steel density were the reinforcement parameters that were studied in the frame of this work. The results demonstrated that the oriented nails can create a reinforcing mesh which efficiently provides a significant flexural strength and ductility to the printed materials.

## 2. Materials and Methods

### 2.1. Materials

The printed mix was made with 30% of cement CEM I 52,5 N by mass (d_50_ = 10 μm, specific density of 3100 kg/m^3^), 10% of limestone powder (d_50_ = 10 μm, specific density of 2600 kg/m^3^), 10% of kaolin clay (d_50_ = 4 μm, specific density of 2600 kg/m^3^) and 50% of 0/1mm river sand of specific density 2700 kg/m^3^.

The mortar mix was first mixed in a Hobart mixer before the addition of water and superplasticizer. The water to cement ratio was (W/C) 0.51. The superplasticizer used was a polycarboxylate-based polymer PCE suspensions containing 20% of dry extract. The percentage of PCE to cement was 0.2%. (by mass). After the addition of water and PCE and a mixing step of 3 min, the mixer was stopped and the bowl was scrapped. The final step consisted in a 1-min high velocity mixing.

The initial yield stress of the samples was measured just after mixing with a vane geometry using the stress growth procedure detailed in [33]. The initial yield stress was equal to 10 kPa. This value was quite high considering cementitious material 3D printing methods and was likely to induce weak bonding at the interface between layer [2]. However, the nails used were 30 mm length and had a diameter of 1.8 mm, and some part of the nails were subjected to a rusting treatment in order to increase their surface roughness by inducing a better interface property with the cementitious matrix (Figure 1).

### 2.2. Samples Geometries and Fabrication

Samples were made by superimposing 10 × 25 mm² rectangular cross section layers of mortar (same section as the one of the nozzle) that were extruded using a screw extrusion system mounted on a WASP 3MT Industrial 4.0 printer (Figure 2). The nozzle head was equipped with a vibrating system to allow for a continuous feeding of the material.

Three-layer samples and ten-layer samples were fabricated and were sawed in 120 mm long samples for the 3-layer samples and in 40 mm long samples for the 10-layer samples as shown in Figure 3.

For the 3-layer samples after the mortar printing of the 3 layers, nails were manually injected in the material using different spacing (10, 20 or 30 mm) and orientation (vertical, 45° inclined toward the sample center or 45° inclined alternatively toward opposite direction) in order to study the reinforcement effect of the nails on the bending strength of the printed samples. Steels densities are computed for each reinforcement configuration and are summarized in Table 1.

For the 10-layer samples, a single spacing was tested (15 mm) and only vertical and crossed configurations were tested. Nails were manually inserted into every two-layer deposit. In order to prevent contact between nails already placed underneath, the nails were slightly shifted (spacing of 18 mm) for their penetration on top of the fourth and eighth layers. This shifting of nails can be seen in Figure 3.

For the real industrial implementation of an automated solution, envisage the use of co-working robots, a concrete printer and an automated nailing machine. The equipment of the printing head with a digitally controlled pneumatic of electric nailing machine could also be a solution.

### 2.3. Mechanical Measurements

In order to assess the reinforcing effect of nailing, the flexural strength of the printed samples was measured using a 3-point bending test after a curing period of 28 days (90% RH, 20 °C). The spam of the bending device was 120 mm and the tests were conducted at a constant displacement rate of 0.1 mm/s. The loading frame was equipped with a force transducer having a 1 N sensitivity.

## 3. Rheological Requirements for the Penetration of Nails

### 3.1. Penetration Theory

The penetration of a needle or a conical penetrometer within a fresh cementitious material has been used in numerous studies in order to study the structural build-up of the material with aging [34,35,36,37].

The penetration resistance of the conical-ended needle like the nails was considered to be equal to the combination of the contribution of the friction of the cementitious material on the conical tip and the contribution of the friction on the circular cross section cylindrical part of the needle that increased linearly with the penetration depth (Equation (1)).
(1)$$F_{pen}=\pi \tau_{0}\left(\frac{Dh_{cone}}{2}+D(h_{pen}-h_{cone})\right)=\tau_{0}\left(\alpha_{bear}+\pi D(h_{pen}-h_{cone})\right)$$
where: *F_pen_* is the penetration force, *τ_0_* is the static shear yield stress of the cementitious material, *h_cone_* is the height of the conical tip of the nail, D is the nail diameter, *h_pen_* is the penetration depth of the nail.

It is possible to generalize (Equation (1)) to all types of tip shape (flat, hemi-cylindrical) using a bearing coefficient *α_bear_* so that the contribution of the tip is equal to *α_bear_τ_0_.* In the conical case, the contribution of the penetration resistance force due to the tip is equal to (Equation (2)):(2)$$\alpha_{bear}\tau_{0}=\frac{\pi Dh_{cone}}{2}\tau_{0}$$

For the sake of simplicity, in the following, the tip contribution is supposed to be punctual and is written using the bearing coefficient.

Note that this equation assumes that the stress at the interface between the nail and the cementitious material was equal to the material static yield stress. This means that the material was sheared at the interface and that slip was not occurring.

### 3.2. Penetration of the Nail within a Printed Sample

In the case of the printed samples that were layered material with layers exhibiting distinct properties, the problem was more complex. The printed materials exhibited a shear yield stress that increased with time at rest, due to structural build-up: the first deposited layers were stronger than the “youngest one”. In the first attempt, the evolution of the static yield stress after deposit can be modeled using a constant rate of increase *A_thix_* as proposed by Roussel et al. [38,39] (Equation (3)).
(3)$$\tau_{0}\left(t_{rest}\right)=\tau_{0,0}+A_{thix}t_{rest}$$
where: *τ_0,0_* is the initial static yield stress (non-structured material), t_rest_ is the resting time.

Note that more complex descriptions of the structural build-up kinetic depend on studied timescales [40,41,42,43]. Therefore, in order to predict the penetration force of the nail within the layered structure, it was necessary to account for the gradient of static yield stress.

In order to illustrate the modeling of the penetration force, the penetration force modeling the printed structure was discretized in *n* layers printed in a period of time *t*. Assuming a constant printing rate *R* (printing height increase), it was possible to link the elapsed time *t* to the printed height *H = n.H_layer_* with *H_layer_* the thickness of a single layer.

Therefore, the printed height since an elapsed period of time *t* can be approximated by *n.H_layer_*/*R.* It was then possible to write the static shear yield stress within the *i^th^* layer from the top printed surface (Equation (4)):(4)$$\tau_{0}\left(i\right)=\tau_{0,0}+A_{thix}\frac{iH_{layer}}{R}$$

Equation (3) allowed us to compute the static yield stress of the material in each layer of the printed structure. By combing Equations (1) and (3), it was possible to obtain the evolution of the penetration force of a nail entirely crossing *n* layers of layered cementitious materials (It was assumed that the nail crossed at least two different layers) (Equation (5)):(5)$$\begin{array}{l} F_{pen}=\alpha_{bear}\left(\tau_{0,0}+(n+1)\frac{A_{thix}H_{layer}}{R}\right)+\sum_{i=1}^{n}\pi DH_{layer}\left(\tau_{0,0}+i\frac{A_{thix}H_{layer}}{R}\right)\\ +\pi D\left(H_{pen}-nH_{layer}\right)\left(\tau_{0,0}+(n+1)\frac{A_{thix}H_{layer}}{R}\right) \end{array}$$

In order to test the modeling of the penetration force, two different types of printing cementitious materials were investigated [44]. The first one corresponds to an accelerated fluid cementitious mortar having a low initial static yield stress (50 Pa) and a high structural build-up rate (500 Pa/min). The second one corresponds to a firm mortar that can be used in the so-called infinite brick method [7], having a high yield stress (3 kPa) with a moderate structural build-up rate (100 Pa/min). The static yield stress evolution of both mortars was plotted on Figure 4. It can be observed that Mortar 1 became stronger than Mortar 2 after 15 min of resting time.

The computed penetration was computed for a printed structure having 10 mm high layers and plotted in Figure 5 for the fluid accelerated mortar (Mortar 1) and in Figure 6 for the firm mortar (Mortar 2) and the nail diameter was supposed to be 3 mm. The penetration force was plotted in function of the penetration depth up to 100 mm, which means that, in the computation, the nailing of up to 10 layers was simulated. Different time gaps ranging from 0 (a theoretical case where no structural build-up of the cementitious materials have occurred) to 15 min. Notably, the maximum aspect ratio of the nail was equal to 33 and the nail could be considered as rigid in front of the cementitious material using the rigidity criterion provided by Martinie et al. [45].

In order to analyse Figure 5 and Figure 6, four scenarios were compared: fast print (short time gap, i.e., below 2 min) and short nails (low penetration depth, for example under 20 mm); slow print and short nails; fast print and long nails (high penetration depth, for example around 100 mm); and slow print and long nails.

In the first case (fast print and short nails), the nails that reinforced the cementitious material did not have the time to strengthen. This means that the penetration force was mostly governed by the initial static yield stress of the material. As a result, the penetration force was close to 0 for the Mortar 1 which had a low initial yield stress and remained under 1 N for the firm Mortar 2.

In the second case (slow print and short nails), the effect of structural build-up was still limited, even in the case of Mortar 1, for a penetration of 20 mm, only two layers were involved and the resting time maximal value is 15 min (maximal static yield stress of 5050 Pa which was almost the same value than the one of Mortar 2 at the same age). The penetration force at 20 mm remained lower than 1 Pa for Mortar 1 and ranges between 2 and 3 Pa for Mortar 2.

In the third case (fast printing and long nails), at 100 mm, the material age was about 10 min. At this time, the static yield stress values of both materials were close. As a result, the penetration force was close and ranged between 2 and 5 N.

In the fourth case (slow printing and long nails), the printing time to print 100 mm was higher than 100 min. At this time, Mortar 1 exhibited higher static yield stress than Mortar 2. As a result, the penetration force in Mortar 1 was about 20 N while it was only 10 N for Mortar 2 when a time gap of 15 min was used.

To conclude, the penetration force was mostly influenced by the static yield stress of the oldest nailed layer. This means that Mortar 1 required a higher penetration force if the shear yield stress of the last nailed layer was higher than that of Mortar 2. This depends on the resting time of the mortar (after 15 min, the threshold of mortar 1 became higher than that of Mortar 2).

It is important to note that the nails were not likely to buckle during penetration. The Euler buckling critical load for a 100 mm long 3 mm in diameter nail is about 800 N, which was far higher than the maximum obtained value of about 20 N. 

It is important to note that the penetration of the needle was possible as long as the material remained plastic. Once the behaviour of the material became frictional, it also began to be fragile, and fracturing could occur during penetration [46].

## 4. Mechanical Reinforcement

### 4.1. Bending Resistance

In order to assess the reinforcement effect of the nails in the different configurations, the maximum bending loads were compared. The results are summarized in Table 2 for bending loads perpendicular to the layer’s direction (three layers sample) and in Table 3 for bending load parallel to the layers direction (10 layers sample).

For the layers tested with perpendicular bending, without reinforcement, the bending resistance was 3250 N and was almost the same with vertical reinforcement with a spacing of 2 and 3 cm and was slightly increased with a spacing of 1 cm. This result was expected because of the nail orientation that was not efficient enough to compensate for the tensile stress induced by bending.

It can be noted that the nail’s reinforcement in the inclined and crossed configurations always increased the maximum bending force. The resistance tended to increase with the nail’s reinforcement density. It can be observed that the maximum recorded force was equal to 4500 N (with rusty nails) for the inclined geometry and 4900 N (with rusty nails) for the crossed geometry for nails with a spacing of 1 cm. In these cases, the maximum force was increased by around 50%. These results show that inclined and crossed nail configurations can be an efficient way to improve the bending strength of printed cementitious composites significantly.

Comparing the results obtained with rusty and smooth nails, it can be observed that the surface roughness had only a limited influence on the maximum bending force, even if average values seemed to be slightly higher with rusty nails.

For the layers oriented parallel to the bending load (10-layer sample), only one spacing was studied (1.5 cm). The results summarized in Table 3 show that nail reinforcement was able to increase by about 50% of the maximum bending force. Neither the surface roughness (smooth and rusty) nor the reinforcement configuration seemed to have a significant effect on the maximum bending load. It is important to note that, in this case, the vertically injected nails with an orientation can efficiently act during bending solicitation. 

It can be noted that the bending resistance was significantly lower for this direction on loading due to the anisotropic behavior of the printed structure. This anisotropic behavior was due to the very high yield stress value of the printed cementitious materials, which led to an imperfect interface between layers [31]. 

### 4.2. Post-Peak Behavior

Concerning concrete construction, steel reinforcement did not increase the bending resistance but provided some ductility to the material in order to avoid a sudden and dramatic collapse of the structure.

In this section, the different parameters of the nailing configuration were compared in order to assess the effect of steel density, steel surface roughness and nail orientation. The study focused on some specific cases (always with bending load perpendicular to the layer) but were representative of the overall observed results.

In Figure 7, the bending load vs. sample deflection curves are plotted for the unreinforced sample, and for reinforced samples using crossed nails spaced by 2 cm with both smooth and rusty surfaces. As shown in Table 2, it can be observed that nails increased the bending resistance and did not depend on the surface roughness. However, the post-peak behavior was different. After the peak of the smooth surface, the load decreased to 0 before increasing to a value close to around 400 N. This behavior was linked to the slippage of the steel nails at the cementitious matrix interface that limited the efficiency of the steel reinforcement. On the other hand, with the rusty nails, there was a dramatic reduction of the force after the peak, however the force did not drop to 0, but to a constant remaining value close to 1000 N. This demonstrated that the interface between nails and mortar must be sufficiently strong to create an efficient reinforcement system.

In Figure 8, the effect of steel reinforcement density on the post-peak behavior was assessed. The bending load vs. sample deflection curves are plotted for the unreinforced sample, and for reinforced samples using crossed rusty nails spaced by 1, 2 and 3 cm. For the 3 cm-spaced nail reinforced sample, after the peak, the remaining force was close to 0. This was due to the fact that there was no nail crossing the mortar cracks in the lower stretched parts of the sample. Comparing spaced nails reinforced by 1 and 2 cm, it can be observed that the remaining force increased with the reinforcement density: it was equal to 1000 N for the 2 cm-spaced nails and around 1300 N for 1 cm-spaced one. This showed that the post-peak remaining force was governed by the reinforcement density.

The results show that the reinforcement density influenced the remaining force after the peak. Additionally, the effect of the nail orientation at constant steel density was investigated. Figure 9 presented the bending load vs. sample deflection curves for the unreinforced sample, and for reinforced samples using constant rusty nails reinforcement density with the three different steel orientation: crossed nails spaced by 2 cm, vertical nails spaced by 1 cm and inclined nails spaced by 1 cm. The results demonstrated that vertical nails were not able to reinforce the sample because they did not cross the cracks of the stretched part of the mortar samples. As a result, there was no remaining force after the peak for the vertical nails. It can be also observed that, for the same reinforcement density, inclined and crossed nails provided the same remaining force after the peak. This result was not so surprising because both configurations had the same numbers of steel nails crossing vertical cracks (with a 45° angle), showing the importance of steel dosage.

### 4.3. Possible Durability Issues and Steel Corrosion

One major drawback of this method is linked to the presence of steel within a cementitious matrix: with time, the cement matrix carbonation process will lead to a reduction of pH or penetration of chloride which will result in steel corrosion. Steel corrosion is a very known pathology of conventional reinforced concrete, and some cautions have to be taken in order to ensure a sufficient service life of the printed structures:

(1) The cementitious materials permeability has to be the lowest possible in order to slow down carbonation process and penetration of corrosive agent. For printed structures, the interface between layers has to be high-quality in order to not be a preferential path of penetration and carbonation.

(2) The cover must be sufficient in order to protect the steel. In this case, covers provided in design codes (and depending on the concrete environment) can be used as a reference value.

(3) Other supplementary materials such as fly ash or ground granulated blast-furnace slag can be used to reduce the corrosion of steel nails.

Another alternative to steel corrosion is to use others types of materials for nails: stainless steels, glass or basalts or carbon can be considered as a potential solution that does not present any risk of corrosion compared to steel nails. 

Using these none-corrosive materials, the requirements for nailing insertions will remain the same as presented in this study, and the effect on the hardened properties is expected to almost be similar.

## 5. Conclusions

In this study, the experimental results show that the proposed nailing reinforcement can exhibit an efficient way to reinforce printed mortar structures. This study demonstrated that the changing rheological behavior of cementitious materials during printing was suitable for the penetration of nails during the process. A simulation of the nailing of up to 10 layers was carried out for two mortars having typical printable behavior. It can be concluded that both the initial yield stress and structural build-up rate had an impact on the penetration force within the layered structure (showing a gradient of strength).

It was also demonstrated that reinforcement, by using nails, was able to efficiently strengthen printed samples if the orientation of the nails was correctly chosen and the nails surface was sufficiently rough to ensure a good interface with the mortar. 

In conclusion, this investigation paved a new path towards fully automated selective steel nail placements as reinforcements during the digital fabrication of concrete in order to strengthen the concrete structure. 

## Figures and Tables

**Figure 1 materials-13-01518-f001:**
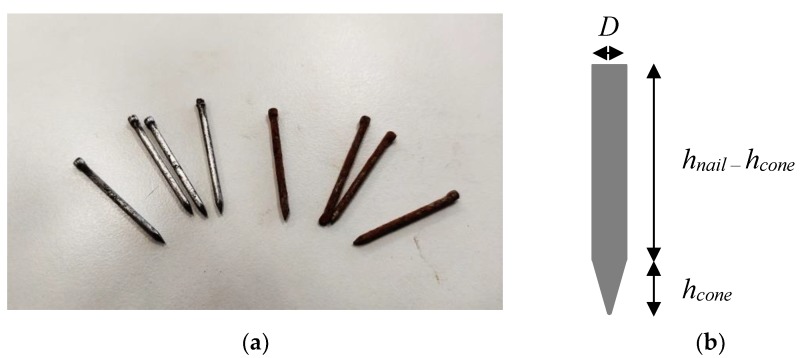
(**a**) Nails before and after rusting treatment; (**b**) Considered nail geometry.

**Figure 2 materials-13-01518-f002:**
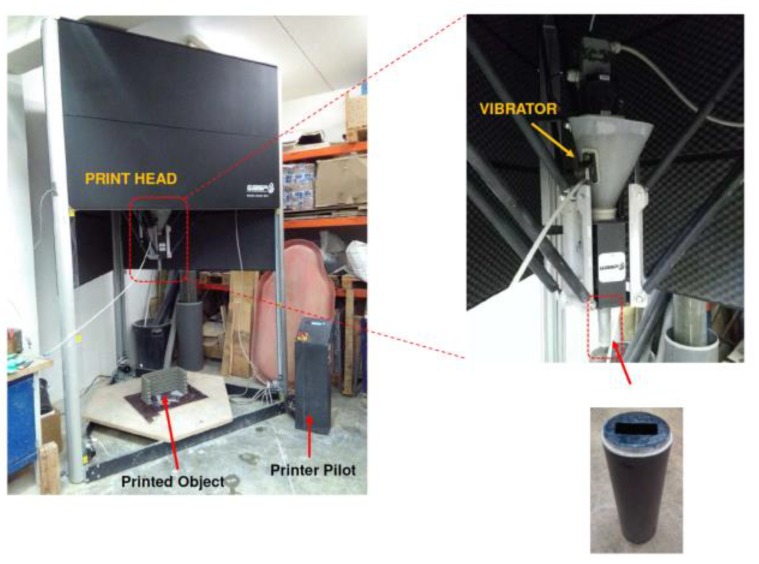
Picture of the printing system: printer, printing head and nozzle.

**Figure 3 materials-13-01518-f003:**
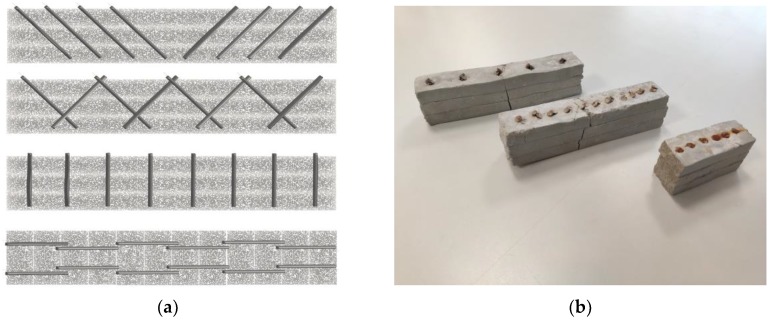
Manufactured samples geometries: (**a**) schematic views; (**b**) pictures of samples after bending tests.

**Figure 4 materials-13-01518-f004:**
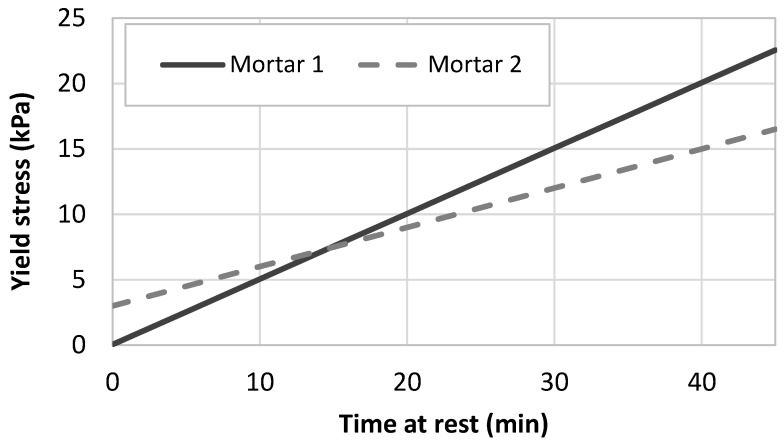
Static yield stress evolution of mortars representative of a fluid accelerated one (Mortar 1) and a firm one used in the infinite brick method (Mortar 2).

**Figure 5 materials-13-01518-f005:**
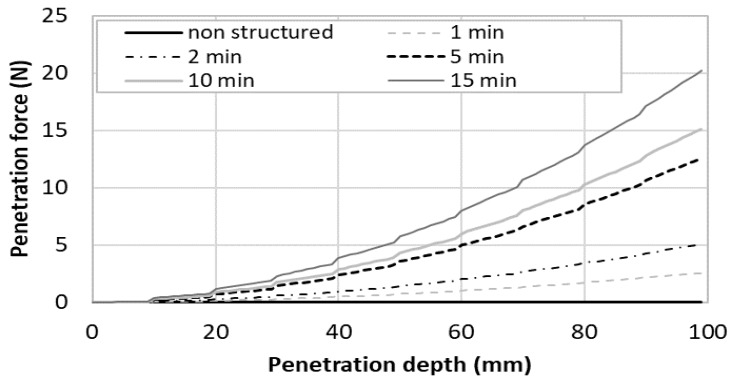
Penetration force evolution of nails (D = 3 mm) within a layered printed structure of Mortar 1 with different time gap ranging from 1 to 15 min.

**Figure 6 materials-13-01518-f006:**
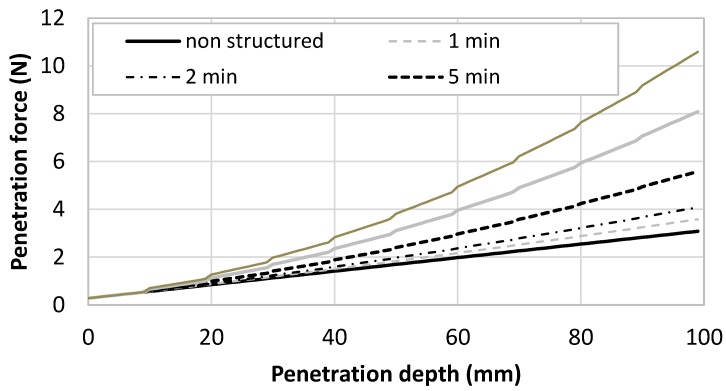
Penetration force evolution of nails (D = 3 mm) within a layered printed structure of Mortar 2 with different time gap ranging from 1 to 15 min.

**Figure 7 materials-13-01518-f007:**
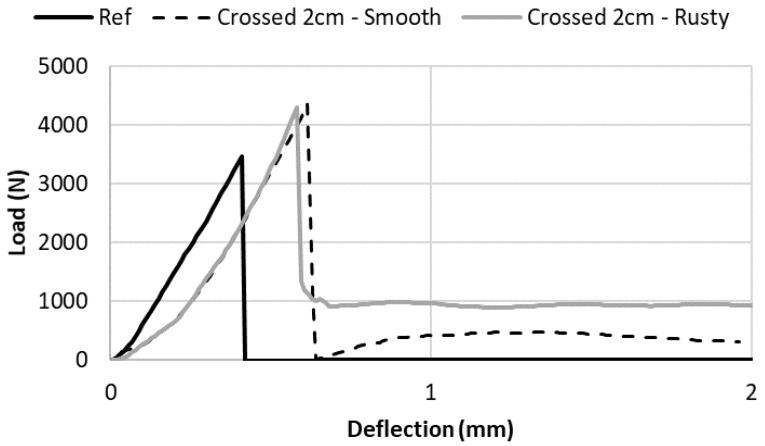
Effect of the surface roughness on the post-peak behavior of the reinforced samples.

**Figure 8 materials-13-01518-f008:**
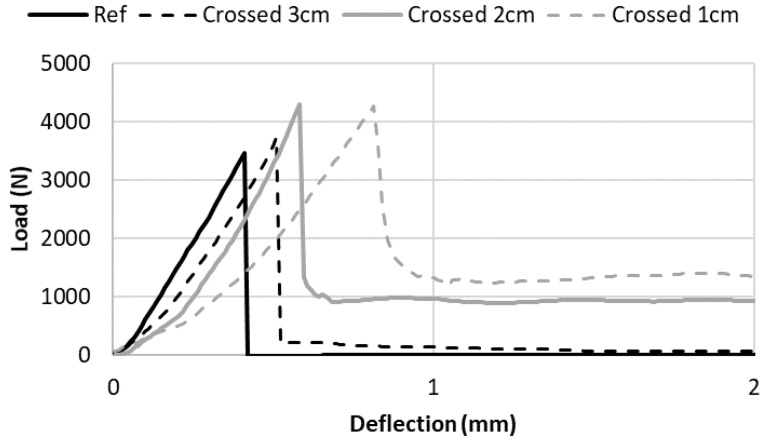
Effect of the density of nails (spacing between nails) on the post-peak behavior of the reinforced samples.

**Figure 9 materials-13-01518-f009:**
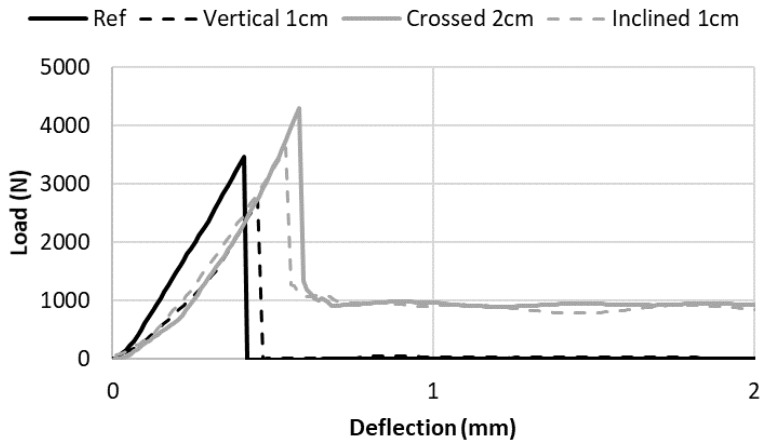
Effect of the nail’s configuration on the post-peak behavior of the reinforced samples.

**Table 1 materials-13-01518-t001:** Reinforcement configuration with nails and steel reinforcement density (in kg per cubic meter of cementitious material.

Reinforcement Geometry	Inclined-Vertical	Crossed	Inclined-Vertical	Crossed	Inclined-Vertical	Crossed
Distance between nails (mm)	30	30	20	20	10	10
Steel weight (kg)/mortar (m^3^)	21	42	32	64	64	127

**Table 2 materials-13-01518-t002:** Maximum bending force recorded for sample with bending force perpendicular to the layer’s direction.

Reinforcement Direction		No	Vertical			Inclined			Crossed		
Distance between Nails (cm)		-	1	2	3	1	2	3	1	2	3
Smooth	Average (N)	3250	3750	3667	3000	4533	4167	4200	4870	4533	3983
	Standard dev. (N)	507	71	416	283	503	321	346	44	115	305
Rusty	Average (N)	3250	3550	3250	3300	4200	4267	3767	4550	4267	3790
	Standard dev. (N)	507	495	495	608	265	702	115	71	321	115

**Table 3 materials-13-01518-t003:** Maximum bending force recorded for sample with bending force parallel to the layer’s direction.

Reinforcement Direction.		No	Vertical	Inclined
Distance between Nails (cm)				
Smooth	Average (N)	640	845	895
	Standard dev. (N)	53	87	42
Rusty	Average (N)	640	974	995
	Standard dev. (N)	53	98	148

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
