# Peer review of "Nailing of Layers: A Promising Way to Reinforce Concrete 3D Printing Structures"

_materials, 2020, doi:10.3390/ma13071518_

Round 1
Reviewer 1 Report
The theme of the 3D printer is certainly topical, but for this reason one should not run the risk of being thrown into hurried studies.
even if I honestly don't fully appreciate the idea of ​​inserting nails into printed objects since the objective with which the 3D printers have been theorized could be lost, the experimental program is well thought out.
I would devote more attention to the aspect of optimizing the mix design.
English and grammar are absolutely to be reviewed as they are often erroneous and inappropriate for a scientific journal.
Have you even tried to think about the problem of corrosion of the nails inserted inside? I believe that a profound reflection and deep-rooted justifications are needed on this aspect
Other comments can be found on the attached PDF

Author Response
The theme of the 3D printer is certainly topical, but for this reason one should not run the risk of being thrown into hurried studies.
even if I honestly don't fully appreciate the idea of ​​inserting nails into printed objects since the objective with which the 3D printers have been theorized could be lost, the experimental program is well thought out.
I would devote more attention to the aspect of optimizing the mix design.
Thank you for the comments from the reviewer.
Authors: This is not the topic of the paper. In this paper the authors focused on the reinforcing effect of nails in printed materials and also on the nails penetration step. This is the reason why in the part related to the penetration step, two different types of materials which are typical of extreme printing conditions were used as explained in Mechtecherine et al. (CCR 2020) and Roussel et al. (CCR 2018). Moreover, the mix design has been the topic of numerous study and we do not want here to address this point.
English and grammar are absolutely to be reviewed as they are often erroneous and inappropriate for a scientific journal.
Authors: The paper has been carefully read and corrected to make the English suitable for scientific journal.
Have you even tried to think about the problem of corrosion of the nails inserted inside? I believe that a profound reflection and deep-rooted justifications are needed on this aspect
Authors: The authors agree that corrosion of steel could be a problem with such technique. The authors add a part on this concern in order to warn the readers on this crucial point.
Other comments can be found on the attached PDF
Authors: Comments have been addressed.
Reviewer 2 Report
Dear Authors,
The article has scientific features and can make a contribution to science, but below I present some comments and suggestions for You.
Thank You for Your Work,
Reviewer
Line 47: „Using fibers was also a common method to reinforce printed concrete structures”
Please write, what kind of fibers for concrete are you talking about? Glass, steel, basalt fibers, etc?
Line 53 and 78: „The aim was to provide ductility, tensile and shear strength” & „The initial yield stress was equal to 10 kPa”
How can adding nails to the raw material in 3D printers improve plasticity of this mass? Please explain the issue carefully.
In my opinion, nails can affect the strength of the final material, but not the mass (to increase plasticity, the Authors write about adding superplasticizers to the raw material - right?).
Point 4.1. – results understandable due to the presence of nails in the material.
The raw material mass (if compared to concrete) must have a certain specific strength (especially compressive and bending strength).
The addition of reinforcement (in various forms) is to strengthen the mass in terms of strength - which is theoretically obvious.
Advanced research is currently underway as part of the possibility of using 3D printing technology in architecture, construction and reverse engineering (currently with high intensity).
Considering the issues and tests presented by the Authors (and hereinafter the results), the title and content of the article should refer to 'material durability' and further the possibilities of increasing the strength of the presented material.
Can Authors attach some photos from the printing process?
The aspect of how nails were placed in the material using the 'layer by layer' printing method may be interesting here.
Conclusion;
Linie 304: A simulation of the nailing of up to 10 layers have been carried out for two mortars having typical printable behavior.
Simulation is a theoretical study aimed at practice, what is a good approach to this topic. But below I make a few comments if the topic relates to practice (in the future).
Line 308-309: „It has been also demonstrated that reinforcement by using nails was able to efficiently strengthen printed samples if nails orientation was correctly chosen and if the nails surface was sufficiently rough to ensure a good interface with the mortar”.
The conclusion is rather clear for a person who knows the specifics of building materials and their properties.
The adhesion of concrete is directly proportional to the degree of roughness of the modifier in physical form (e.g. nails).
Similarly, old concrete when combined with fresh concrete should also be partly forged for a good connection.
In my opinion, the article lacks images, e.g. from the Scanning Electron Microscope (SEM) to establish the connection on the "mass/nail"border.
Similarly, tests on the microstructure of building materials (concrete as well) are carried out using SEM to check the phase arrangement (calcium silicates hydrated) or the contact zone aggregate/SiO2/pores/binder/phase.
I suggest that the Authors supplement the content with microscopic examination (it can also be an optical microscope).
These studies are not complicated, and photographs will definitely be a variety and supplement to the content of the presented tests and this paper.
In addition, on the following pages:
-http://www.totalkustom.com/
-https://newstorycharity.org/
-https://www.dezeen.com/2019/07/24/3d-printed-concrete-choreography-pillars-design/
-Modern Ornamental (so-called: digital carving, printing, hardening and fixing of elements).
In these links you can also find information about: printing houses in 3D technology and concrete columns (ETH Zurich).
It is interesting and technologically excellent, as printed materials already have adequate strength and reduce working time.
In addition, please provide:
- what type of apparatus was included in printing?
- for what purpose/target market/volume or surface of building objects the materials described in the article would be dedicated (in more detail than specified).
Literature extensive in relation to the entire text.
Thank You.
Author Response
The article has scientific features and can make a contribution to science, but below I present some comments and suggestions for You.
Thank You for Your Work.
Thank you for your comments.
Reviewer
Line 47: „Using fibers was also a common method to reinforce printed concrete structures”
Please write, what kind of fibers for concrete are you talking about? Glass, steel, basalt fibers, etc?
Authors: The detail was provided in the manuscript:” Using fibers was also a common method to reinforce printed concrete structures [12,13,25–29]. For example, steel [25], bio-based [16,27], polymeric [12,25], basalt [29] or glass fibers [14] have already been tested to enhance the tensile behavior and the ductility of 3D printed cementitious materials.”
Line 53 and 78: „The aim was to provide ductility, tensile and shear strength” & „The initial yield stress was equal to 10 kPa”
How can adding nails to the raw material in 3D printers improve plasticity of this mass? Please explain the issue carefully.
In my opinion, nails can affect the strength of the final material, but not the mass (to increase plasticity, the Authors write about adding superplasticizers to the raw material - right?).
Authors: Nails was inserted after the deposit step. Therefore, it does not change the fresh state but only the final hardened properties of the printed materials. The authors revise and improve the writing of this part to make it clearer. “In this paper, a new strategy of reinforcement was investigated. This strategy consisted in the placement of nails through several layers after their deposit. The aim was to provide ductility, tensile and shear strength while giving additional strength to the interface between layers once the material hardened. This strategy can be easily automated using a robotic placement of the nail which can be a real advantage and beneficial in the context of digital construction.”
Point 4.1. – results understandable due to the presence of nails in the material.
The raw material mass (if compared to concrete) must have a certain specific strength (especially compressive and bending strength).
The addition of reinforcement (in various forms) is to strengthen the mass in terms of strength - which is theoretically obvious.
Advanced research is currently underway as part of the possibility of using 3D printing technology in architecture, construction and reverse engineering (currently with high intensity).
Considering the issues and tests presented by the Authors (and hereinafter the results), the title and content of the article should refer to 'material durability' and further the possibilities of increasing the strength of the presented material.
Authors: To address the aspect of durability, the authors add some sentences in order to warn the readers on the risk of corrosion and propose other alterantives.
Can Authors attach some photos from the printing process?
Authors: The picture of the printer has been added in Figure 2.
The aspect of how nails were placed in the material using the 'layer by layer' printing method may be interesting here.
Authors: The nails had been manually inserted. Therefore, there is no photo of that. Nevertheless, the authors add more details on how it was carried out.
Conclusion;
Linie 304: A simulation of the nailing of up to 10 layers have been carried out for two mortars having typical printable behavior.
Simulation is a theoretical study aimed at practice, what is a good approach to this topic. But below I make a few comments if the topic relates to practice (in the future).
Line 308-309: „It has been also demonstrated that reinforcement by using nails was able to efficiently strengthen printed samples if nails orientation was correctly chosen and if the nails surface was sufficiently rough to ensure a good interface with the mortar”.
The conclusion is rather clear for a person who knows the specifics of building materials and their properties.
The adhesion of concrete is directly proportional to the degree of roughness of the modifier in physical form (e.g. nails).
Similarly, old concrete when combined with fresh concrete should also be partly forged for a good connection.
In my opinion, the article lacks images, e.g. from the Scanning Electron Microscope (SEM) to establish the connection on the "mass/nail"border.
Similarly, tests on the microstructure of building materials (concrete as well) are carried out using SEM to check the phase arrangement (calcium silicates hydrated) or the contact zone aggregate/SiO2/pores/binder/phase.
I suggest that the Authors supplement the content with microscopic examination (it can also be an optical microscope).
These studies are not complicated, and photographs will definitely be a variety and supplement to the content of the presented tests and this paper.
Authors: We agree with the reviewer that it can have been of interest to provide a microstructural analysis even if it is not the main topic of the paper (it is more a proof of concept of a new reinforcing approach). However, due to covid-19 crisis, university research lab in France is closed for weeks and we will not be able to do these experiments right now. It can be the topic of further studies.
In addition, on the following pages:
-http://www.totalkustom.com/
-https://newstorycharity.org/
-https://www.dezeen.com/2019/07/24/3d-printed-concrete-choreography-pillars-design/
-Modern Ornamental (so-called: digital carving, printing, hardening and fixing of elements).
In these links you can also find information about: printing houses in 3D technology and concrete columns (ETH Zurich).
It is interesting and technologically excellent, as printed materials already have adequate strength and reduce working time.
Authors: Thank you for providing these interesting links.
In addition, please provide:
- what type of apparatus was included in printing?
Authors: We use a WASP 3D printer MT. It is mentioned in the manuscript.
- for what purpose/target market/volume or surface of building objects the materials described in the article would be dedicated (in more detail than specified).
Authors: This is not our purpose right now. The study is not market driven. The authors want to investigate the possibility of improving the ductility of printed cementitious composites.
Literature extensive in relation to the entire text.
Thank You.
Reviewer 3 Report
The paper brings an original proposition how to deal with one of the crucial barriers for concrete 3D printing, namely a lack of the sufficient bonding between the printed layers of concrete. The presented aproach is quite simple - using nails - yet promising and definitely worth to consider. In my opinion the paper is wel prepared and can be published in the journal Materials with some minor additions:
- there is stated in the Section 2.2 that for the 3-layers samples nails were manually injected in the material. What about the 10-layers samples? Please deccribe the nailing procedure in more detailed way;
- following the above, please consider and comment the possibilities of automation of this process from the practical point of view. How it could be organized on-site?
- the use of rusted nails could be favourable for the temporary adhesion between layers, but in longer periods the issue of durability has to be taken into consideration. Please consider and comment.
Author Response
Reviewer 3
The paper brings an original proposition how to deal with one of the crucial barriers for concrete 3D printing, namely a lack of the sufficient bonding between the printed layers of concrete. The presented approach is quite simple - using nails - yet promising and definitely worth to consider. In my opinion the paper is well prepared and can be published in the journal Materials with some minor additions:
- there is stated in the Section 2.2 that for the 3-layers samples nails were manually injected in the material. What about the 10-layers samples? Please describe the nailing procedure in more detailed way;
Authors: Additional information has been provided: For the 10-layers samples, a single spacing was tested (15 mm) and only vertical and crossed configurations are tested. Nails were manually inserted every two layer deposits. In order to prevent the contacts between nails already placed underneath, nails are slightly shifted (spacing of 18 mm) for their penetration on top of the fourth and eighth layers. This shifting can be seen in Figure 3.
- following the above, please consider and comment the possibilities of automation of this process from the practical point of view. How it could be organized on-site?
Authors: The possibility of using co-working robots or equipping the printing head with a digitally controlled pneumatic or electric nailing machine is evocated.
- the use of rusted nails could be favorable for the temporary adhesion between layers, but in longer periods the issue of durability has to be taken into consideration. Please consider and comment.
Authors: A new part dealing with durability has been added to warn the reader on this crucial point.
Round 2
Reviewer 1 Report
Accept. All the comments and suggestions have been addressed.